# Farmers’ Views and Tools Compared with Laboratory Evaluations of Parasites of Meat Goats in French West Indies

**DOI:** 10.3390/ani13030422

**Published:** 2023-01-26

**Authors:** Jacques Cabaret, Mélodie Mercier, Maurice Mahieu, Gisèle Alexandre

**Affiliations:** 1UMR 1282, Infectiologie et Santé Publique (ISP), INRAE and Tours University, Bât. 213 Centre de Recherches Val de Loire, 37380 Nouzilly, France; 2INRAE, UR0143 ASSET Agroécologie, Génétique et Systèmes d’Élevage Tropicaux, Domaine Duclos, 97170 Petit-Bourg Prise d’eau, France

**Keywords:** gastrointestinal nematodes, parasites, goats, farmers’ views

## Abstract

**Simple Summary:**

Parasites are a major health problem in tropical goat husbandry. Among the parasites, gastrointestinal nematodes (GINs) play an important role. The control of GINs relies on the use of synthetic anthelmintics. Their use for decades has resulted in frequent GIN resistance to these treatments. Therefore, there is a need to target only the animals in need of treatment to reduce the selection for GIN resistance. The difficulty is in finding easy to use pathophysiological indicators such as anemia (with FAMACHA©, estimation of eye mucosal color), loss of weight (with body score), or hair roughness. These indicators can be used by farmers after the promotion and help of agricultural advisory services. Laboratory examinations such as the excretion of GIN eggs in feces are very useful in estimating the infection of goats, but are costly and thus remain the exception. Eighteen farmers participated in semi-directive interviews to appreciate their relation to goat GIN infection and the solutions they considered. Seventeen were investigated for GIN infection. The average infection by GINs was high, with a wide range from one farm to another. The frequency of anthelmintic treatments was negatively related to the use of body score, FAMACHA©, and hair roughness. GIN were not a major issue for traditional farmers. This is due to the important use of indicators and the belief in their value, which provides comfort to farmers that the parasites are being controlled.

**Abstract:**

Gastrointestinal nematodes (GINs) are a major health problem in tropical goat husbandry. The control of GIN has been nearly exclusively reliant on the use of anthelmintic treatments. Their wide use has provoked the appearance and diffusion of anthelmintic resistance. Therefore, there is a need to use anthelmintics only when they are really needed. This strategy of targeted selective treatment (TST) has been recommended. The selection of animals to be treated has been based either on the objective measures of GIN intensity (fecal nematode egg counts) performed in the laboratory or on indirect assessment such as anemia (FAMACHA©), diarrhea score or weight gains, particularly in sheep. The roughness of hair has also been proposed in goats. These indicators can be handled by the farmer. Their opinion on the importance of GINs, and the indicators that they are ready to accept and use have very rarely been studied. Goat for meat production is important in the French West Indies (especially in Guadeloupe) and GIN infection may significantly alter this production. Eighteen farmers participated in semi-directive interviews in order to appreciate their relation to goat GIN infection and the solutions they considered. Seventeen farms were investigated for fecal nematode egg counts, FAMACHA©, body score, and roughness of hair. The average infection by GINs was high (average fecal egg count 1562 and standard deviation 2028) with a wide range from one farm to another (from 0 to 25,000 eggs of GIN per gram of feces). The *Haemonchus* genera was predominant (54%), followed by *Trichostrongylus* (37%) and *Oesophagostomum* (9%). Young goats were less infected than adult goats since they were not yet grazing; males were more infected than females; and the Creole breed was more infected than the other breeds. Among the farming types, the professional ones were less infected compared with the traditional or mixed agriculture and husbandry farms. Those using targeted selective treatment did not have a significantly higher GIN infection than those treating the whole herd. Most of the characteristics were related and multivariate analysis could not match the intensity of GIN infection with any parameter. The frequency of anthelmintic treatments was negatively related to the use of body score, FAMACHA©, and hair roughness. The use of semi-directive interviews provided a wider understanding of the strategies and problems of farmers. The farmers valued their animals very much and diseases, in general, were a preoccupation, whereas parasites were not a major issue for traditional farmers. This is due to the important use of indicators and the belief in their value that gives comfort to the farmers that the parasites are being controlled. The extension services have well diffused the practice of indicators to the goat farmers of Guadeloupe, with some depending less on anthelmintics to control the gastrointestinal nematodes by using targeted selective treatments.

## 1. Introduction

Farmers’ knowledge is based on different aspects, from books and schools (algorithmic), from someone (mimetic) and from what you feel (phoric) [1]. In French sheep farming for meat, algorithmic and mimetic learnings are the main ones [2]. Concerning animal health, the farmers’ decisions are influenced by veterinarians or technicians [3], even if these stakeholders hold different views [4]. The parasitic infection of goats has a negative impact on their production in European conditions [5] as well as in the Caribbean [6]. The endoparasitic infections in Guadeloupe (French West Indies) are frequent and gastrointestinal nematodes (GINs) (mostly *Haemonchus contortus* and *Trichostrongylus* sp. [7]) are a major concern. Farmers cannot easily detect the presence and intensity of GIN infection. The objective evaluation is mostly based on nematode fecal egg counts provided by the laboratory. This may be costly and time-consuming to collect and transport the samples to the laboratory. There is also some time delay between the collection of fecal samples and the laboratory results. This is why some tools easily available by the farmers themselves have been proposed. The farmer’s eye score is relatively good to detect groups of animals showing poor performances, but these are not always related to GIN infections [3]. Some other pathophysiological scores may be of interest to detect sheep with high infection: the dag-score for *Trichostrongylus* [8], diarrhea score [9], weight decrease [10], body condition score [11], and anemia in *H. contortus* infection [12,13]. FAMACHA© is a system designed to identify anemia using the eye mucosal color: the correlations with anemia (correlation r ranged from 0.35 to 0.68) are quite good [14]. The correlations (r ranged from 0.13 to 0.36) were much weaker with infection based on the fecal nematode egg counts [14]. However, FAMACHA© remains one of the most widely used system in tropical regions where infection with *H. contortus* is widespread. It has also been shown that the absence of roughness of hair is indicative of fair productivity in goats [15]. The FAMACHA© or its derivate (estimating roughly the eye mucosal color), diarrhea, and roughness of hair has been proposed to farmers in Guadeloupe either by a cooperative or/and the local INRAE (French National Research Institute in Agronomy and Environment) research center for more than 15 years. We will estimate if the use of these indicators has become widely used by farmers. Anthelmintic drugs have been extensively used to control GINs and has resulted in the appearance of the resistance of GIN in most countries and in Guadeloupe [16,17]. To reduce the spread of anthelmintic resistance, targeted selective treatments (TST) have been proposed to restrict the use of anthelmintics to the animals that are most infected [18] with some success. It has been also applied in Guadeloupe [6] and in other regions in tropical goats [19,20]. This was sometimes based on FAMACHA© alone or associated with body scores [19]. The use of all of these indicators requires the motivation of the farmer, which is based on the perceived susceptibility and severity of GINs, the benefits of using indicators and barriers, and finally, cues to action. The Health Belief Model [21] was originally established to understand how humans make medical and health care decisions and focuses on the threat perception and behavioral evaluation of health problems. This model can be readily extended to any kind of decision to be taken in veterinary medicine, particularly to GIN infection, which does not always cause clear symptoms. Thus, certain GIN control practices will be attractive depending on the farmer’s beliefs relative to the appreciation of their animals, the role of GINs, and the expected returns from these practices. The farmers may not comply with all of the technical advice [22], but their health practices are not irrational [23]. Their way of thinking about diseases is quite different from that of the veterinarian [24], and has rarely been investigated, although this is very important if one wants to propose actions to regulate diseases. Goats for meat in the French West Indies, particularly in Guadeloupe, are economically important and have cultural and religious implications for the” Indian” community [25,26]. This community is constituted of the offspring (fourth generation) of immigrants from India; they represent 10% of the Guadeloupe population. We will investigate the perceived importance of goats and their well-being, the role of GINs on production, and their acceptance of tools (FAMACHA©, body condition score, roughness of hair) to decide on treatments among meat goat breeders; we will also evaluate the capacity of these tools to predict the actual parasitic infection.

## 2. Materials and Methods

### 2.1. Description of the Farms

Eighteen farms (coded from F0 to F17) were surveyed for semi-directive interviews and 17 of them were investigated for parasites. An interview of one technician from the cooperative (T) was also performed. Ten farms were in Grande Terre and seven in Basse Terre (n = 7). Basse Terre is a volcanic mountainous area, and the rainfall ranges from 2000 to 4000 mm. Grande Terre is calcareous, with a flat landscape and the rainfall is less than 1500 mm. There are three farming types [27]: agriculture and husbandry (n = 3), professional husbandry (n = 4), and traditional husbandry (n = 11). The latter corresponds to amateur breeders also practicing another professional activity not related to husbandry. The farming types of Galan et al. [27] corresponded to those in Fanchone et al. [28]. The size of the herd comprised between 10 and 90 goats; the main breeds are Anglo-Nubian and Boer, Creole, and Creole mixed breeds. The area of the farms extends from less than one hectare to 10 ha. Most of the farms have pastures and pen (n = 11), some have only pastures (n = 3), and others breed their goats only inside a pen (n = 4).

### 2.2. Farmer Semi-Directive Interviews

A short visit of the farm was conducted before the interviews. The interviewers (M.Me and J.C. or M.Ma.) asked the farmers open questions, as described by [29]. These questions had been pre-prepared in an interview guide, which was identical for all interviews (Appendix A). The interview guide for the cooperative technician was adapted from the farmers’ guide. The recorded interviews were transcribed into a Word text. Tropes (V8.5) [30] speech analysis software was first used to process the data for the cognitive analysis of the interviews [31], which were then analyzed using the multivariate method [32] applied to the most frequently used words in the interview. Significant differences in the word’s occurrence between farming types were assessed using Z score statistics for two populations; where the proportions were low (less than 4%), Fisher’s exact test was applied to the number of occurrences for each word. The classical way to interpret interviews is also used based on the most exemplary sentences. Technical questions on GIN management (type of anthelmintic treatment, frequency, and pasture use) were asked at the end of the semi-directive interview.

### 2.3. Farm Evaluation of Gastro-Intestinal Parasites

Feces was collected from the goats that were not treated recently, although this time-lag from treatment could vary from one farm to another and may alter the interpretation of fecal GIN egg counts (FEC) expressed in GIN eggs per gram of feces (EPG). The MAFF modified McMaster method [33] was used to determine the EPG. Briefly, sodium chloride as a flotation solution (specific gravity 1.18) was used; the sensitivity of 50 EPG (nematode eggs per gram of feces). The presence/absence of *Moniezia* eggs and coccidian oocysts was noted. After fecal culture, GINs were identified for each farm after seven days of culturing, based on the identification keys in [34].

### 2.4. Pathophysiological Indicators

The animals were randomly selected irrespective of age, sex, and breed for the fecal egg counts of parasites, FAMACHA©, body condition, and hair appearance. The FAMACHA© system is based on a semi quantitative evaluation of the eye mucosal color. This color was classified into one of five categories according to the FAMACHA© eye color chart: 1 = red, non-anemic; 2 = red-pink, non-anemic; 3 = pink, mildly anemic; 4 = pink-white, anemic; 5 = white, severely anemic [12]. The scale of assessment of the body condition ranged from 0 to 5. The values inferior to 1 indicate a need of treatment (in sheep: [35] and in goats [36]). The hair appearance was graded from A to D (A = shiny, B = normal, C= slightly spiky D = spiky and/or dull) as already evaluated in goats by [37] in Guadeloupe and in [15] in Italy.

### 2.5. Statistical Analyses of Characteristics of Farms

The relationship between EPG and farm or farmer management characteristics were conducted by one-way analysis of variance, ANOVA (qualitative data), or with non-parametric Spearman correlation for the quantitative ones, with the SPSS 11.5 software. The EPG were Napierian log transformed prior to analysis due to their aggregated distribution. Cluster analysis based on centroid grouping and Gower distance (variables being binary, multistate, or quantitative) was conducted with MVSP software. Cluster analysis permitted relating all the variables at once.

## 3. Results

During farm visits, their structures (pen and pastures) and herds were rated as poor (one farm), average (one farm), good (11 farms), or very good (four farms).

### 3.1. Intensity of Infection by Endoparasites in Relation to Farm Characteristics

The average EPG of gastrointestinal nematodes was 1562 and the standard deviation 2028 with a wide range (min = 0 and max = 25,000).

The *Haemonchus* genera was predominant (54% of the identified larvae), followed by *Trichostrongylus* (37% of larvae) and *O. oesophagostomum* (9% of larvae), based on the average of the farms’ fecal cultures. The GIN fecal egg counts and farm characteristics are presented in Table 1. Internal parasitism was variable according to farms (*p* < 0.001) but also to animal age. Kids (< 3 months; n = 18) were less infested by strongyles (*p* = 0.02), equally infested by *Moniezia* (*p* = 0.35), and more by coccidia (*p* = 0.07) than the adults (n = 141). Males (n = 43) were more infected by GINs than females (n = 98; *p* = 0.04). 

The characteristics of the farmers and farms were related to GIN infection. Animals from farms with pasture and pen were more infected than those with only pasture or no pasture. The farming system was related to the infection level (*p* < 0.001). Thus, animals from the “professional husbandry” group were less infected than those from the “agriculture and husbandry” or “traditional husbandry” groups. Belonging to a husbandry organization was also related to significantly higher GIN infection. The farmers that were new in husbandry had herds with less infections (*p* = 0.01).

Among the 17 farmers interviewed, nine treated the entire herd and eight used targeted selective treatment mostly based on FAMACHA (Table 2). Targeted selective treatment was more widely used on large herds (average 48 goats and standard deviation 24) and the treatment of all animals in smaller herds (average (17 goats and standard deviation 5). The method of treatment as well as their frequency does not appear to be directly connected to herd infection.

### 3.2. Infection Relation with All the Characteristics of the Farm

Since most of the characteristics of farms are interrelated, we used a cluster analysis of relationships between these characteristics and the helminth infection on the individual values of 156 goats (Figure 1). First, it can be noticed that FEC (expressed in eggs of GIN per gram of feces) was not positively related to any of the farm characteristics (Gower similarity = −0.25). Second, many characteristics were highly related (Gower > 0.90): area and age of the farm (older farms accumulated land), age and sex of goats (the males are sent earlier to abattoirs), relation between pertaining to a husbandry organization and goat breed (cooperative promoting Creole breed), frequency of anthelmintic treatments and body score (recommended by agricultural extension services), hair appearance, and FAMACHA© (highly recommended by extension services), opinion on parasitism and health importance (parasitism considered as a very important negative component of health), the use of anthelmintic related to farming system (more targeted treatment in professional husbandry with higher number of goats), association between the presence of *Moniezia* and *Coccidia* (probably associated in young goats), the use of pasture more frequent in farmers having parents already in husbandry (possibly due to the accumulation of land along with time). Some groups were less homogeneous (Gower = 0.5): size of flock, area and age of the farm (larger land may support more goats); age, sex, breed and belonging or not to husbandry organizations (Boer breed presented larger carcasses than the Creole-promoted by the cooperative and one breeder association, thus reducing the percentage of older male kids in the herd); the frequency of anthelmintic treatments related to all pathophysiological indicators available for farmers; treatment method (all herd or targeted) related to farming system and opinion on health and parasitism of the farmer; other internal parasites (*Moniezia* and *Coccidia*) related to the presence of pasture and parents already involved in husbandry (possibly due to the accumulation of land and pasture and thus infection with *Moniezia*).

### 3.3. Farmers’ Tools to Evaluate Internal Parasites

These were evaluated only in adult goats since few kids were investigated. FAMACHA© was distributed among categories 1, 2, and 3 (68, 47, and 21 goats, respectively); none of these corresponded to real anemia (categories 4 or 5) and this indicator was not related to EPG (ANOVA, *p* = 0.54). The body scores of the goats were 5, 4, 3, and 2 for 46, 46, 11, and two goats, respectively, and none corresponded to the necessity of treatment (score 1 or 0); this indicator was not correlated to EPG (ANOVA, *p* = 0.15). The combination of the three indicators was not also correlated to EPG (ANOVA, *p* = 0.39). The hair coat appearance was graded into 47 A, 69 B, 20 C, and one goat with D. It was rarely spiky (D), and this indicator was not related to EPG (ANOVA, *p* = 0.79).

### 3.4. Farmers’ Opinion on the Value of Animals

All of the farmers had a high opinion of their animals, and they really cared: “It is the passion I already have for animals, especially for goats” (F1)), “We always have this habit, when they are very small, we always pick them up like that, cuddle them” (F8). For some of them, they considered their animals as a part of their own well-being “as I say often, it helps me to… to escape a little bit” (F10), “..animals… Their company… I find they bring me a lot… from a personal point of view too” (F17). For many of the traditional farmers, the economic value of husbandry was not the main issue: “So I am going to do it for fun, not really to make money” (F5), “We do it more for pleasure than for money” (F11). Even for those oriented toward economic return, they did not consider their animals as simple machines for production “You must produce…, But I say to myself, well, if they like to run around a bit, well, let them run around. They will lose a bit of energy, but it is fine.” (F4).

### 3.5. Farmers’ Problems Related to Husbandry

Some were related to the structure of the farm and society. A common one was the difficulty of feeding the goats (six out of 18 farms). Nine farms had one hectare or less, and the surface was often also dedicated to another activity (market gardening or food crops). Thus, it was due to the very limited surface available for each farm, which did not allow to grass to be produced sufficiently. “The farms are very small. We will have to change the model, we should have farms with at least 20 to 30 ha” (F0). The possibility of irrigation was not always available due to a lack of water, and this also reduced the production of fodders (three out of 18 farms). The second limitation was financial: the farmers could not always afford to buy complementary feed. The damage caused by stray dogs was also recorded in 13 out 18 farms. Six farms out of 18 mentioned the stealing of goats. Other problems were related to pathology (ecthyma, cowdria, ticks, gastrointestinal helminths, and coccidia in four, two, five, five, and two farms, respectively). 

The analysis of word occurrence (Table 3) was conducted based on traditional (those having an extra income from another occupation) and professional/agriculture and husbandry farmers (they obtained income from their farm). The regrouping of the last two was due to the many similarities (larger areas, larger herds, and entrepreneurial attitude). As expected, the word goat was the most frequently found for all farmers. The traditional farmers less felt the impact of season on goat production (since their feed was obtained from external sources or their own banana leaves for a limited number of goats), they were more reliant on drugs (other than anthelmintics) in problem solving health, they were not using the words parasite and anthelmintic, and they favored the use of plants when available and vitamins. They had more complaints on water availability mostly for animal and pasture use. All of the farmers mentioned hair appearance and eye mucosal color equally.

The use of Tropes for the recorded interviews allowed for a multidimensional analysis presented in Figure 2. Traditional husbandry had few words integrated (8) in the speech instead of 15 in the professional one, with more details on pathology events (ticks, scabies, coccidia, helminth, resulting eventually in mortality). They also complained more about dog attacks and robbery, and were preoccupied by their production efficiency. Traditional husbandry farmers have a more unclear vision of disease and health, rely strongly on the veterinarian to solve diseases, and they are interested to obtaining advice; they are also in favor of using plants to treat animals. *Aloe vera* and sea water were among the most known remedies, mostly for gastro-intestinal problems, among all of the farmers.

Computerized analyses of speech have their limits and a careful reading of interviews and the extraction of sentences of interest may also shed light on information not detected otherwise. The workload and constraint of husbandry, financial shortage for a more professional development, and the efficacy of anthelmintics were the main aspects detected by the extraction of sentences of interest. The workload and constraints were mentioned only in four farms, even if it was very difficult for some of them: “No, it is days, some days you’re…you’re much more tired, some weeks too.” (F17), “, … in the evening I’m worn out, I can tell you.” (F2), “You have to… not lose yourself in this job which, … which eats up a bit, which draws, which draws a lot of energy.” (F4). Financial shortage was partly associated with the difficulty to increase land (as seen in Figure 2 for traditional husbandry),”No, but we have other problems to get the land. It wasn’t easy” (F13), but it also includes buildings etc.: “equipment, things… these are big investments.” (F11). It also takes time before you obtain results from investments: “… you realize that the bank account is more in the red, because it requires investments, and in fact these returns on investment are not immediate.” (F6). The efficacy of anthelmintics was considered on seven farms: “There are some…molecules that don’t hold up anymore. The fact of constantly giving… so you must be able to vary the molecules. And everything you can buy can be used once or twice, then it’s chaos.” (F0). The statements on anthelmintic resistance and the shortage of land and cash were confirmed by the cooperative technician: “The professional would like to evolve but he cannot evolve because he has no land, he has no agricultural water, because he has no cash flow.”, “.., they have the information. But the practical application is quite different because of the limitations on land, water, and cash”. She also acknowledged the problem of anthelmintic resistance: “We realized that worms were 100% fenbendazole resistant, so there was no point in delivering it”.

## 4. Discussion

Farmers’ tools such as eye mucosal color and hair appearance were widely used and well-considered due to the extension work made by the cooperative, INRAE, and some local veterinarians. As classified by [1], the learning was partly through leaflets and other documents provided by the cooperative, and mostly by learning from technicians, researchers, veterinarians, or family members. The use of body score was measured in the present work, but requires more time and expertise than the other indicators and was not part of the farmer’s knowledge. The Health Belief Model [21] provides a framework to understand the success of indirect indicators: goats are considered as valuable, the GIN infection is mainly due to *H. contortus,* which causes anemia and loss of weight that can also be evaluated by the farmers, and these indicators of GIN infection are easily implemented. The value of goats as a production animal has been recognized, as seen from previous surveys [25]; the emotional value (it is a way of life to have goats) is also important, as seen in the present survey. However their value was not viewed as that important by the agriculture and husbandry farmers, where the main effort and income was on agriculture. There were several differences between professional and traditional farmers (Figure 2). The importance of helminths (and GINs) was more acknowledged by professional farmers than by traditional ones. The latter were less specific on their diseases, and they relied much more on veterinarians and their anthelmintics. The evaluation of hair appearance and the eye mucosal color are conducted based on the recommendations of agricultural extension services; it is not always carried out with a clear grid of values as we did in this survey. Additionally, we did not check if the indicators were always performed, as recommended by the promoters of the method [12]. The mention of humidity by farmers (traditional ones, Figure 2) and the management of grazing when available was probably derived from agricultural extension services and based on research in Australia [38] and local data [39]. The existence of GIN resistance is also known by several farmers due to extension information based on research results [16,17]. The alternative use of anthelmintic with different modes of action is also known, even if the multiple brand names for the same molecule might confuse farmers. The targeted selective use of anthelmintics [6], as proposed elsewhere [18,40], based on indirect indicators, was also employed by some farmers in Guadeloupe, mostly those with larger herds: it significantly reduced the cost of anthelmintics and provided a lower selection of resistant GIN. There were no significant differences in the EPG of the farmers using targeted selective treatment or all herd treatment. The decisions of the farmers incorporated the knowledge from extension services and were not irrational, as already pointed out [23].

The EPG may be considered as indicative of GIN infection, but their value may be altered by the existence of a treatment one or two months ago, and it cannot be considered as a completely exact measure of GIN infection. However, some data in Table 1 agree with the known features: young animals before weaning are less infected than adults, and males are more infected than females (Table 1). Even without pasture, the goats were infected, which was not apparently expected. It should be, however, borne in mind that indoor animals are given fresh grass from formerly grazed pastures and thus may be infected with GINs. A surprising fact was that Creole goats were apparently more infected than Boer or Anglo-Nubian, which are considered by most Guadeloupe farmers as better producers with lower resistance to diseases and GINs. The Boer goats were evaluated as less resistant than local breeds [41] in Texas, but the variability in Creole resistance is very high [42], so it is difficult to conclude whether the recorded EPG difference was due to the breed or environmental factors. Finally, the EPG can probably be taken as a rough estimate of infection in our case, as it corresponded to known environmental and animal characteristics. The most surprising result was the negative and low relationship with farm characteristics, as shown in Table 1. Farm characteristics are diachronic: they are relatively stable and are the result of the history of the farm (size of the farm, type of farming, farmer education and practices, etc.). The EPG of GINs is synchronic: it is quite dependent on the moment of data collection (recent treatment, existence of GIN resistance to anthelmintics, etc.). The low negative relationship between EPG and farm characteristics could then be due to the structure of the data (e.g., synchronic versus diachronic).

It is obvious that goat husbandry is not only facing GIN problems. The small size of farms and the difficulty to increase it due to land shortage is one of the major problems. The small size of farms and the lack of accounting documents may also limit the chances of obtaining European subsidies and thus increase the financial difficulties. However, GINs and other parasites are considered by farmers as a real problem. Although they have incorporated the main information on GINs in their practice, some improvement can still be made, first to evaluate the indirect indicators more accurately, and second, the use of EPG for a more objective evaluation of GINs. The use of fecal egg counts was used exceptionally in Guadeloupe, and was more used in metropolitan France: 31% of sheep farmers use fecal egg count at least once a year [2]. The cost of fecal egg counts could be a limitation of their use in Guadeloupe and cheaper alternatives to individual sampling and evaluation such as composite sampling of the herd [43] could be promoted.

## 5. Conclusions

Indirect indicators of gastrointestinal parasites have been proposed to farmers and tested in natural or experimental conditions as well as in the field. The detailed opinion of farmers on the use of indicators has not been frequently recorded, although it is of utmost importance if one wants to have farmers use these indicators. The agricultural extension services have well promoted the indicators and their meaning to the goat farmers of Guadeloupe, and one third of them is now depending less on anthelmintics to control gastrointestinal nematodes by using targeted selective treatments. Refining and standardizing the practice of indirect indicators as well as using gastrointestinal fecal egg counts on a regular basis would be progress in achieving the control of these parasites.

## Figures and Tables

**Figure 1 animals-13-00422-f001:**
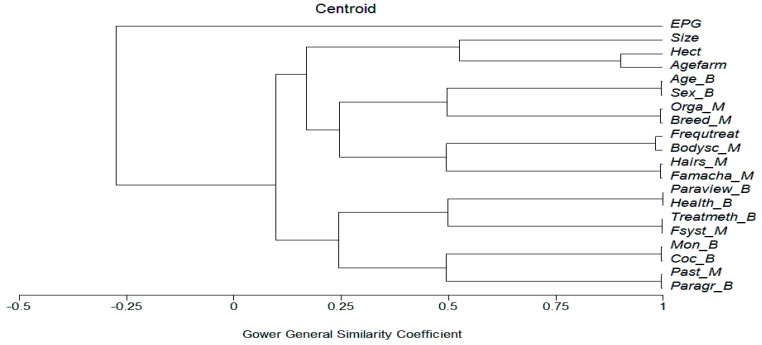
Cluster analysis of the relationship between infestation and the farm characteristics: Quantitative, Binary (B), Multistate (M), nematode eggs per gram of feces (EPG), size of the herd (Size), surface area of the farms in hectares (Hect), age of the farms (Agefarm), age of goats—adult or kids (Age), sex of goats—male or female, farm organization—no pasture, pasture and pen, only pasture (Orga), (Breed), frequency of treatment (Freqtreat), body score condition (Bodysc)—0 very poor to 5 very good, FAMACHA© (Famacha)—very good, good, average, bad, very bad, hair appearance (Hair)—shiny, normal, spiky and dull, farmer’s view on parasites (Paraview) and health (Health), treatment method—all or target selective treatment (Treatmeth), farming system (Fsyst)—professional husbandry, agriculture and husbandry, traditional husbandry, husbandry organization (Orga)—breeder association, cooperative, independent, Moniezia—presence, absence (Mon), coccidian—presence, absence (Coc), use of pasture (Past)—yes or no, farmer parents involved in agriculture—yes or no (Paragr).

**Figure 2 animals-13-00422-f002:**
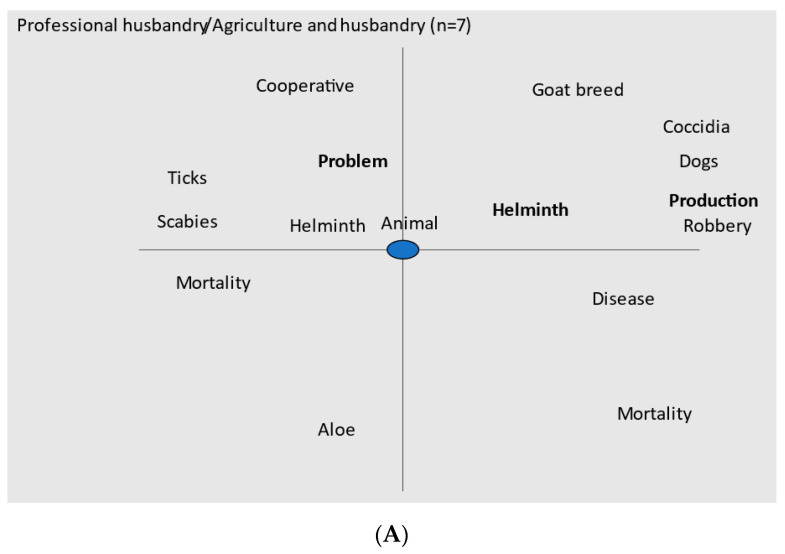
Qualitative content analyses of interviews of farmers practicing professional or traditional husbandry using Tropes software. The analyses were centered on the word animal (or goat); the left part of the figure corresponds to the words appearing before the central word and the left part to the words appearing after the central word; the distance between the central word and other words corresponds to the intensity of the relationship, strong if near or weak when distant. (**A**: Professional husbandry/ Agriculture and husbandry) (**B**: Traditional husbandry).

**Table 1 animals-13-00422-t001:** Characteristics of the farms and their relationship with fecal egg counts of gastrointestinal nematodes. Fecal egg counts are related to the qualitative data (ANOVA) or quantitative data (Spearman r).

Characteristics	Fecal Egg Counts (FEC) or Relationship with FEC (Spearman r)	Significance (ANOVA or r)
***Region*** (Basse-Terre, Grande-Terre)	1490, 1329 ^1^	NS
** *Goats* **		
Age (young vs. adult goat)	540, 1502	*p* < 0.001
Breed (Anglo-Nubian and Boer, Creole, crossbred Creole, and others)	1190, 3751, 1055	*p* = 0.01
Sex (male vs. female)	1691, 829	*p* = 0.04
Presence of sheep (6% of farms) or not	709, 1675	NS
** *Farm* **		
Age of the farm in years (range: 1–37)	r = 0.13	NS
Size of herd (range 11–90)	r = 0.10	NS
Pasture (No pasture, Pasture with a pen, Pasture only)	508, 1746, 843	*p* = 0.03
** *Farmer* **		
Formerly trained (25%) or not in husbandry	2837,1288	NS
Parents were farmers or not	1636, 1135	*p* = 0.01
Farming type (agriculture and husbandry, professional husbandry, traditional husbandry)	2633, 320, 1393	*p* < 0.001
Husbandry organization (Breeder association, Cooperative, Independent)	1206, 1724, 799	*p* = 0.05
** *Parasite management with anthelmintics* **		
Type of treatment (targeted selective treatment vs. all herd)	1919, 935	NS
Frequency of treatment per year (2–6)	r = −0.02	NS

^1^ FEC 1490 in Basse-Terre and 1390 in Grande-Terre

**Table 2 animals-13-00422-t002:** Anthelmintic treatment management by farmers and fecal egg counts (FEC) of gastrointestinal nematodes. N.A., not available.

Farm Code	Average FEC	Standard Deviation	Anthelmintic Treatment	Number Per Year
F1	1574	1108	Targeted	2
F2	4938	5388	*N.A.*	*N.A.*
F3	745	1278	*N.A.*	*N.A.*
F4	99	103	Targeted	Variable
F5	1932	1989	All herd	4.5
F6	501	619	*N.A.*	6.0
F7	48	60	Targeted	Variable
F8	2562	741	All herd	2
F9	7907	8967	Targeted	Variable
F11	28	35	All herd	3
F10	843	1383	All herd	2.5
F12	605	682	Targeted	Variable
F13	936	1367	All herd	2.5
F14	85	90	All herd	3
F15	899	1577	Targeted	Variable
F16	1490	1887	All herd	2
F17	1364	3352	*N.A.*	*N.A.*

**Table 3 animals-13-00422-t003:** The percent occurrence of words used by different categories of farmers.

Words Employed (in Percent)	Professional and Agriculture/Husbandry Farmers (n = 7)	Traditional Farmers (n = 11)	Significance (Chi-Square or Exact-Fisher Tests)
Goat	19.3	12.1	NS
Disease	16.5	11.4	NS
Season	16.5	8.9	S
Problem	11.1	12.7	NS
Drug	7.3	13.4	S
Hair appearance	6.4	7.0	NS
Eye mucosal color	6.4	4.6	NS
Parasite	5.5	0	S
Veterinary surgeon	5.5	12.1	NS
Anthelmintic	5.5	0	S
Plant use	0	7.6	S
Vitamins	0	5.1	S
Water availability	0	5.1	S

## Data Availability

Available on requirement.

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
