# Peer review of "Farmers’ Views and Tools Compared with Laboratory Evaluations of Parasites of Meat Goats in French West Indies"

_animals, 2023, doi:10.3390/ani13030422_

Round 1

Reviewer 1 Report

This paper describes the results of an interview survey carried out on different kinds of goat farmers in Guadalupe. The results are interesting, relevant to the field, and warrant publication. Sometimes, the paper is difficult to understand, and I have drawn the editors and authors attention to some of these sections below. One of the most important messages I took from the paper is that these farmers are informed about the best parasite control practices but are limited in their ability to enact them due to “the limitations on land, water, and cash”. This suggests education alone is not enough to encourage farmers towards best parasite control practices. They appear to need infrastructural supports also. 

The title is confusing and doesn’t fully capture what the paper is about.

Line 16/17: Please define extension services first as it is not clear what it means to the reader.

Line 18: Eighteen farmers were interviewed, and 17 were submitted for parasitological screening?

Line 59/60: I’m not familiar with the terms algorithmic, mimetic and phoric being used to describe knowledge acquisition. I would use the words in brackets instead of those terms to prevent confusion for the reader.

Line 79: unclear  

Line 107: I’m not sure about Guadalupe but “native” is usually seen as a more politically correct term to use than “Indian” which could be considered offensive in the USA for example.

Line 121: The farming types of [27] were corresponding to those of [28] – please add text to this sentence so it makes sense. e.g. The farming types in the study carried out by Galan et al [27] corresponded to those reported by Fanchone et al [28]

Line 141: Faeces was collected from goats

Line 150: Delete “taken”

Line 150: Change “race” to “breed”

Line 225: Change “chara cseveralteristic” to “characteristic”

Line 236-238: Its not clear what the authors are trying to say here.

Figure 2: Needs a caption.

Line 328/329: algorithmic and mimetic are terms I have never seen used in this context. I would reword to avoid confusion

Author Response

The title is confusing and doesn’t fully capture what the paper is about.

We modified slightly the title.

Line 16/17: Please define extension services first as it is not clear what it means to the reader.

Extension services replaced by Agricultural advisory services

Line 18: Eighteen farmers were interviewed, and 17 were submitted for parasitological screening?

Yes, modified in simple summary. Also changed in the abstract.

Line 59/60: I’m not familiar with the terms algorithmic, mimetic and phoric being used to describe knowledge acquisition. I would use the words in brackets instead of those terms to prevent confusion for the reader.

Line 79: unclear 

The sentence has been modified.

Line 107: I’m not sure about Guadalupe but “native” is usually seen as a more politically correct term to use than “Indian” which could be considered offensive in the USA for example.

One sentence has been added to explain who these Indians are. They are called in Creole z’indiens or Chappé coolies. The Indians in Guadeloupe are not native. They are the offsprings of migrants from India (Calcutta and Pondichery); they were introduced from 1850 for 30 years to replace the liberated slaves in the sugar cane plantations. They belonged to the lowest Indian caste and that is why their interpretation of hindouism is different, including sacrifice of goats.

Line 121: The farming types of [27] were corresponding to those of [28] – please add text to this sentence so it makes sense. e.g. The farming types in the study carried out by Galan et al [27] corresponded to those reported by Fanchone et al [28]

Corrected

Line 141: Faeces was collected from goats

Corrected

Line 150: Delete “taken”

Corrected

Line 150: Change “race” to “breed”

Corrected

Line 225: Change “chara cseveralteristic” to “characteristic”

Corrected

Line 236-238: Its not clear what the authors are trying to say here.

Modified: the different scores or codes are attributed to the number of goats having a specific score (in brackets) .

Figure 2: Needs a caption.

A caption is added.

Line 328/329: algorithmic and mimetic are terms I have never seen used in this context. I would reword to avoid confusion

Modified.

Reviewer 2 Report

I have added many comments to the pdf file. Please check all of them. 

Author Response

I have followed the majority of comments

I made answers in your pdf file of evaluation.

Round 2

Reviewer 2 Report

Not all the suggestions I made in the first review have been performed. In fact, I'm marking again some questions in the v2 version. The lack for dispersive mesurements in some points is critical, and the use of literal citations from conversations with farmers/technicians is a bit abusive in some sections.
